# Quantum Models of Consciousness from a Quantum Information Science Perspective

**DOI:** 10.3390/e27030243

**Published:** 2025-02-26

**Authors:** Lea Gassab, Onur Pusuluk, Marco Cattaneo, Özgür E. Müstecaplıoğlu

**Affiliations:** 1Department of Physics, Koç University, Istanbul 34450, Turkey; omustecap@ku.edu.tr; 2Department of Biology, University of Waterloo, Waterloo, ON N2L 3G1, Canada; 3Faculty of Engineering and Natural Sciences, Kadir Has University, Istanbul 34083, Turkey; 4QTF Centre of Excellence, Department of Physics, University of Helsinki, P.O. Box 43, FI-00014 Helsinki, Finland; marco.cattaneo@helsinki.fi; 5TÜBİTAK Research Institute for Fundamental Sciences, Gebze 41470, Turkey; 6Faculty of Engineering and Natural Sciences, Sabanci University, Istanbul 34956, Turkey

**Keywords:** quantum consciousness, Posner model, quantum entanglement

## Abstract

This perspective explores various quantum models of consciousness from the viewpoint of quantum information science, offering potential ideas and insights. The models under consideration can be categorized into three distinct groups based on the level at which quantum mechanics might operate within the brain: those suggesting that consciousness arises from electron delocalization within microtubules inside neurons, those proposing it emerges from the electromagnetic field surrounding the entire neural network, and those positing it originates from the interactions between individual neurons governed by neurotransmitter molecules. Our focus is particularly on the Posner model of cognition, for which we provide preliminary calculations on the preservation of entanglement of phosphate molecules within the geometric structure of Posner clusters. These findings provide valuable insights into how quantum information theory can enhance our understanding of brain functions.

## 1. Introduction

The prevailing assumption in both modern science and philosophy is that consciousness arises from complex synaptic computations within neural networks, where brain neurons function as fundamental units of information [1,2]. However, a purely algorithmic and deterministic perspective seems to leave little room for the inclusion of concepts such as qualia and free will in the understanding of consciousness. Consequently, the term “quantum” has become a popularly used prefix in fields like social science [3] and integrative neuroscience [4], even though the connection between quantum phenomena and consciousness remains a subject of ongoing debate within the physics community [5]. Is the brain really acting as a quantum computer? Have we figured out each and every process in the brain, irrespective of whether it is classical or potentially quantum? There are quite a few terms, such as mind [6,7,8], consciousness [9,10], and instincts, pertaining to the brain or what goes on inside it, that are not accurately defined because we do not have the appropriate tools yet to gauge them [11], and hence to understand them from a physical perspective.

Quantum theory has been employed to explore various aspects of brain activity in fascinating ways [12,13]. Each existing model focuses on a specific dimension of neural processes, such as consciousness, cognition, or perception (including numerosity perception; see, for instance, the spin model in [14]), with the overarching goal of understanding the brain. Our objective is not to prove or disprove any theory or to provide a definition distinguishing consciousness from cognition. Instead, we will use the terminology employed by the authors of these theories. If a theory uses the term “consciousness,” we will use it as well.

Additionally, this perspective paper does not offer a comprehensive review of all models available in the literature. While we acknowledge the significance of well-established approaches, such as the dissipative quantum model of the brain [15] and the holographic brain theory [16,17], these fall beyond the scope of our discussion. Instead, our focus is on three specific models that explore the potential influence of quantum effects on mental processes at three distinct levels: (i) the electrons within neurons, (ii) the electromagnetic fields surrounding neurons, and (iii) the molecules that mediate neuronal communication.

These models include the orchestrated objective reduction (Orch OR) theory [18,19,20,21,22], which suggests that the collective states of electrons inside neurons may function as qubits, with their objective and orchestrated collapse mediated by microtubule molecules playing a key role in the emergence of consciousness; the conscious electromagnetic information (CEMI) field theory [23,24,25,26,27], which predicts that the electromagnetic field enveloping the neural network can interact with individual cells via single photons, potentially enabling analog quantum computation; and the Posner model of cognition [28], which explores a molecular form of quantum computation that employs resources such as quantum entanglement between nuclear spins to synchronize individual neurons.

Our aim is to provide a perspective from the standpoint of quantum information and demonstrate how it can aid in investigating the aforementioned theories, placing particular emphasis on the Posner model of cognition. Additionally, we will present simulation results based on our existing toy model [29], focusing on different geometrical clusters that preserve entanglement and comprise the tetrahedral geometry, which is characteristic of phosphate in Posner molecules. Our numerical results demonstrate that this specific geometry not only better preserves coherence but also maintains entanglement. To better understand our results and the effects in various geometric configurations, we represent the geometry by diagonalizing the Hamiltonian, a method that clearly illustrates the impact of buffer isolation on quantum information protection. These findings offer valuable insights into how quantum information theory can enhance our understanding of brain function.

The paper is organized as follows. Section 2 examines the role of microtubules in consciousness. Section 3 investigates the concept of consciousness in the electromagnetic field. Section 4 focuses on the Posner model of cognition. Building on this concept, Section 5 presents our study on the preservation of entanglement within different geometrical clusters. Section 6 offers a theoretical explanation of our results. Finally, Section 7 concludes the review by summarizing the key findings and discussing their broader implications for future research.

## 2. Quantum Consciousness Emerging from the Microtubules Within Neurons

The foundational assumption of the Orch OR theory [19,22,30], originally introduced by Nobel laureate Roger Penrose in Refs. [31,32], is that consciousness arises from a sequence of discrete events beyond the scope of any computable process. According to this theory, these events may result from a specific quantum phenomenon known as Penrose objective reduction of the quantum state. An alternative objective reduction model proposed by Diósi suggests the emission of radiation during the collapse process. This prediction, however, has been experimentally refuted by Donadi et al. [33]. In contrast, Penrose’s model, which does not involve spontaneous radiation emission, remains a plausible theoretical framework. The objective reduction phenomenon, as proposed by Penrose [34,35,36], involves the collapse of the quantum state in a manner that is not solely the result of environmental interactions but is inherently linked to the fundamental influence of space-time on quantum superposition. When these ideas were combined with anesthesiologist Stuart Hameroff’s research on the biomolecular information processing models [37,38], the concept of orchestrated objective reduction was integrated with biological systems [18,19,20,21], further advancing the development of the Orch OR theory [22]. Consequently, the Orch OR theory is also referred to as the Penrose–Hameroff model of consciousness.

The brain is a warm and noisy environment, which is generally not very conducive to quantum effects. For Penrose’s idea to be plausible, quantum effects would need to survive in the brain at least until a neuron has managed to fire. Hameroff suggested that microtubules, which are large polymers essential to cellular structure and function, could provide a potential framework for quantum processes within neurons. These hollow tubes, composed of tubulin dimers, contain π electrons that delocalize within the aromatic rings of tubulin molecules. In its early formulations [18,19,20,21], the Orch OR theory posited that tubulins could exist in a superposition of distinct mechanical conformations, influenced by London force dipoles within these aromatic rings. However, more recent iterations of the theory have shifted focus away from conformational superpositions, emphasizing instead the superposition of tubulin excitation dipole states as the primary mechanism for information encoding [22]. Both in the initial proposals and in more advanced interpretations of the model, it is hypothesized that π electrons may become delocalized to such an extent that they could form a network potentially capable of sustaining quantum superposition long enough to influence conscious thought [39,40,41,42,43,44,45].

A key challenge for the Orch OR theory is whether quantum coherence can persist in the brain’s warm and noisy environment. Tegmark [46] estimated that decoherence in microtubules would occur in the order of $10^{-13}$ s, making quantum effects seemingly irrelevant. However, Hagan, Hameroff, and Tuszynski [47] recalculated decoherence times, accounting for dielectric properties, dipole interactions, and quantum shielding effects. Their revised estimates extended coherence times to $10^{-5}$–$10^{-40}$ s, which aligns with neurophysiological processes. This time scale is crucial because it suggests that quantum coherence may be sustained long enough to influence neural processing and could play a role in the brain’s larger-scale functions.

While the Orch OR theory presents a framework for linking large-scale quantum effects to brain activity, there remains a significant conceptual gap between these bottom-up approaches and the higher-level aspects of consciousness, such as cognition, perception, and self-awareness. The transition from microtubule-level quantum phenomena to macroscopic brain function is not yet well understood, and the connection to psychological or philosophical interpretations of consciousness remains speculative. This gap also manifests in the difficulty of directly correlating quantum events with observable cognitive states. For instance, EEG signal drops—often associated with altered states of consciousness—may be linked to the collapse of quantum superpositions within the brain’s microtubules, reflecting the loss of coherence and the cessation of certain quantum processes that could contribute to consciousness. To place these theoretical achievements into perspective, further discussion is needed on how quantum effects—if present—could integrate with known neuroscientific and cognitive mechanisms. This issue highlights the broader challenge of bridging physics-based models of consciousness with the complexities of human experience and cognition.

### Superradiant Excitonic States in Microtubules

Recent experimental studies have demonstrated that microtubules can capture and transfer energy over distances of approximately 6.6 nm, surpassing classical predictions [48,49,50]. This quantum phenomenon suggests that energy transfer in biological systems may adhere to quantum mechanical principles rather than classical pathways [51,52]. However, the specific mechanisms underlying this energy transfer remain unknown.

A promising investigation into the potential for long-range coherent quantum phenomena in cytoskeletal microtubules involves the study of superradiance [39,40]. This quantum phenomenon occurs when molecular sites interact with a shared electromagnetic field, leading to collective light emission as excitations become delocalized across multiple molecules simultaneously.

Exciton delocalization plays an important role in biological networks of photoactive molecules by providing giant transition dipoles that can strongly couple to the electromagnetic field, enable superabsorbtion and supertransfer, and transport the cellular photoexcitation to a specific reaction center. The molecular structures and their quantum dynamics need to be modelled as open systems, where the first level of the environment is the electromagnetic field to which the “exciton waves” couple. Traditional quantum optics assumes that photoactive molecules are identical two-level systems and models the delocalization of a single excitation and its transfer back to the electromagnetic field through photon emission by an effective non-Hermitian Hamiltonian [53,54,55,56]:(1)$$H_{\mathrm{eff}}=H_{0}+\Delta -\frac{i}{2}G\;,$$
whose right and left eigenvectors do not form an orthogonal basis individually, but instead establish a biorthogonal basis collectively, a defining feature of non-Hermitian quantum systems [57,58,59,60]. More precisely, given that $H_{\mathrm{eff}}|E^{R}\rangle =E|E^{R}\rangle$ and $\langle E^{L}|H_{\mathrm{eff}}=E\langle E^{L}|$, the eigenvectors satisfy the biorthogonality condition $\langle {E^{\prime }}^{L}|E^{R}\rangle =c_{E}\;\delta_{E^{\prime },E}$, leading to the spectral decomposition $H_{\mathrm{eff}}=\sum_{E}E|E^{R}\rangle \langle E^{L}|/c_{E}$.

In Equation (Equation 1), $H_{0}$ accounts for the on-site energies of a single excitation in the molecular aggregate, and Δ and *G* describe the coupling of the exciton between different sites as a result of the interaction with the electromagnetic field. This non-Hermitian Hamiltonian offers an opportunity to go beyond the dipole–dipole approximation, which is a limiting case of the interactions described by Equation (Equation 1) when the size of the system is much smaller than the wavelength associated with the transition dipole of the molecular sites [54].

The non-Hermitian Hamiltonian approach outlined above has been recently utilized in Ref. [61] to develop a theoretical model for exciton transport within microtubules. In this model, aromatic tryptophan amino acids (Trp)—that have the largest transition dipole moment in the structure of microtubules—were treated as the photoactive molecular sites through which a single exciton delocalizes. The positions, dipole orientations, and excitation energies of these molecules were obtained by previous molecular dynamic simulations and quantum chemistry calculations. Then, the biorthogonal eigenspectrum $\{E=E-i\Gamma /2,|E^{R}\rangle ,|E^{L}\rangle \}$ of Equation (Equation 1) was obtained by diagonalizing it for a system consisting of up to 100 spirals of 13 microtubule protofilaments. To this aim, each spiral was assumed to include 104 Trp dipoles. Figure 1 presents a schematic representation of the tryptophan network in the microtubule.

The imaginary part Γ of the complex eigenvalues E represents the decay width that determines the coupling of the extended excitonic state with the electromagnetic field. When the decay widths of all right eigenstates $|E^{R}\rangle$ were plotted after being normalized by the single dipole decay width γ in [61], it was realized that the superradiant state corresponds to the lowest excitonic state for the systems that have more than 12 spirals. It was also demonstrated that either full or a partial randomization of Trp dipole orientations destroys the superradiance, which in turn indicates that the superradiance depends on the particular order of these dipoles in microtubules.

The extent of exciton delocalization was also investigated for both the superradiant and subradiant states of 100 spirals in [61]. To this aim, the authors utilized the three dimensional visualization of the probability of finding the exciton on Trp site *k*, which reads(2)$$P\left(k\right)=\frac{|\langle k|E^{R}\rangle {|}^{2}}{\sum_{k}{\left|\langle k|E^{R}\rangle \right|}^{2}}\;.$$

This investigation revealed that the exciton in the superradiant state is delocalized over all Trp sites on the microtubule’s external wall, potentially facilitating communication with cellular proteins. Conversely, exciton delocalization in the long-lived subradiant state is concentrated on the inner wall of the microtubule lumen, possibly contributing to neuronal process synchronization. The same probability-based analysis using smaller microtubule segments (specific groups of 13 coupled spirals, the minimum number ensuring the superradiant state as the lowest excitonic state) showed that the ground state of the whole segment acts as a coherent superposition of the ground states of its smaller components. Furthermore, photoexcitation was found to spread ballistically along the longitudinal axis, and the superradiance demonstrated robustness to disorder, even with uniformly distributed excitation energy across Trp dipole sites.

This type of probabilistic analysis of exciton delocalization within a microtubule segment could be further refined by incorporating established measures from quantum information theory. For instance, assume the microtubule system reaches thermal equilibrium at inverse temperature β, then, provided that the values of E are real or come in complex conjugate pairs with the same degeneracy [57,58,59,60], the state of the system is given by(3)$$\rho_{\mathrm{th}}=\sum_{E}\frac{e^{-\beta E}}{Z}|E^{R}\rangle \langle E^{L}|,$$
where $Z=\sum_{E}e^{-\beta E}$. The probability of finding the exciton on Trp site *k* turns out to be(4)$$P\left(k\right)=\langle k|\rho_{\mathrm{th}}|k\rangle =\sum_{E}\frac{e^{-\beta E}}{Z}\langle k|E^{R}\rangle \langle E^{L}|k\rangle ,$$
which can be nonzero across the microtubule segment, although its state is an incoherent mixture. In this example, the simultaneous presence of the exciton at two different Trp sites with nonzero probabilities cannot be interpreted as the exciton being delocalized across these two sites. While P(k) and P(j) may have positive values, the off-diagonal term of $\rho_{\mathrm{th}}$, which couples these two sites, $\langle k|\rho_{\mathrm{th}}|j\rangle$, can still be zero. Thus, probability-based approaches might be misleading in witnessing exciton delocalization, especially when the microtubule is an open system.

Alternatively, the density matrix formalism provides a powerful framework for quantifying exciton delocalization in microtubules using suitable measures of quantum coherence $C_{m}$ [62]. This methodology parallels the approach used to quantify proton delocalization in water ice systems, as detailed in Ref. [63]. Furthermore, measures of quantum superposition [64,65] can be applied when 〈k|j〉≠0, drawing an analogy to the analysis of electron delocalization in aromatic molecules, as described in Ref. [66].

Quantum coherence measures the degree of quantum superposition in a state ρ with respect to an orthogonal basis {|k〉}. One widely used measure is the $l_{1}$ norm of coherence, defined as(5)$$C_{l_{1}}\left[\rho \right]=\sum_{k\neq k^{\prime }}\left|\langle k|\rho \right|k^{\prime }\rangle |.$$

The $l_{1}$ norm of coherence is maximized for an excitonic state ρ=|ψ〉〈ψ| where$$|\psi \rangle =\sum_{k=1}^{N}\frac{1}{\sqrt{N}}|k\rangle .$$
This corresponds to a state where the exciton can be found in each Trp site with equal probability 1/N upon measurement. Conversely, $C_{l_{1}}$ vanishes for incoherent mixtures like $\rho_{\mathrm{mix}}=\sum_{k}P\left(k\right)|k\rangle \langle k|$. Another measure of coherence is the relative entropy of coherence [62]:(6)$$C_{RE}\left[\rho \right]=\min_{\sigma \in I(S)}S\left(\rho ||\sigma \right)=S\left(\rho_{d}\right)-S\left(\rho \right),$$
where the minimization is over the set of incoherent states in the basis {|k〉}, S(ρ||σ) is the quantum relative entropy, and $S\left(\rho \right)=-\mathrm{tr}\left(\rho \log_{2}\rho \right)$. $S(\rho_{d})$ is the von Neumann entropy of the state $\rho_{d}=\sum_{k}\langle k|\rho |k\rangle |k\rangle \langle k|$. The relative entropy of coherence measures how distinguishable a density matrix is from its closest incoherent state, providing a way to quantify the coherence present in the system.

This quantum information-theoretical approach also allows us to rigorously measure exciton delocalization over two blocks of 13 coupled spirals, say blocks *A* and *B*. By taking a partial trace over the degrees of freedom of the other blocks $\bar{X}$, we can calculate the reduced state $\rho_{X}$ for any block or block cluster X={A,B,AB}. Then, $C_{m}\left[\rho_{A}\right]$ and $C_{m}\left[\rho_{B}\right]$ quantify intra-block exciton delocalization inside blocks *A* and *B*, respectively. The quantity $C_{m}\left[\rho_{AB}\right]-C_{m}\left[\rho_{A}\right]-C_{m}\left[\rho_{B}\right]$ corresponds to inter-block exciton delocalization, related to the coupling between blocks. This approach provides a theoretical background for experiments observing exciton delocalization via interference in emission lines from spatially separated blocks. An experiment similar to the one performed for polyacetylene in [67] can be designed for microtubules.

In addition to enhancing the quantification of delocalization, the perspective of quantum information science could provide an opportunity to extend the non-Hermitian Hamiltonian approach to a quantum master equation approach. This would allow for modeling the complete dynamics of photoexcitation in microtubules, including the influence of the surrounding environment [53,54,55,56]. The effective Hamiltonian $H_{\mathrm{eff}}$ given in Equation (Equation 1), which characterizes the reduced photoexcitation dynamics, can be derived from a couple of extended master equations that include the electromagnetic field’s degrees of freedom [53,54,55,56]. However, these master equations should also be extended to include thermal fluctuations of the surrounding environment.

The non-Hermitian approach introduced in Ref. [61] and discussed above, which is relevant to the single-excitation manifold, was recently extended in a follow-up study to investigate the effects of superradiance induced by ultraviolet excitation of several biologically relevant Trp mega-networks [48]. In this study, the researchers enhanced their theoretical and computational results with experimental fluorescence quantum yield measurements in tubulin and microtubules. Interestingly, they observed that the formation of strongly superradiant states—resulting from collective interactions among over $10^{5}$ Trp UV-excited transition dipoles within microtubule structures—enhances the efficiency of ultraviolet light absorption and its redistribution to lower energy levels. This mechanism may provide potential protection for cells against ultraviolet-induced damage.

The insights from quantum information science that we have explored to enhance the theoretical framework in Ref. [61] are equally applicable to the more recent study in Ref. [48], offering a unified perspective on exciton delocalization and superradiance in microtubules. In particular, quantum coherence and entanglement measures could provide a more rigorous quantification of exciton delocalization, while a master equation approach incorporating environmental interactions may offer deeper insight into the robustness of superradiant states under physiological conditions. For instance, quantifying superradiant state formation using the $l_{1}$ norm of coherence or relative entropy of coherence could reveal the extent of quantum delocalization, while entanglement entropy between different microtubule segments may provide insights into the role of quantum correlations in exciton transport. Furthermore, incorporating thermal fluctuations, vibrational coupling with the surrounding protein matrix, and solvent-induced decoherence into a master equation framework could provide a more realistic description of microtubule photoexcitation dynamics in physiological environments. Such an approach could also help elucidate whether superradiance-mediated exciton transport plays a role in intracellular signaling or energy redistribution pathways. Experimental verification through time-resolved spectroscopy or modified fluorescence quenching assays may further substantiate these theoretical predictions.

## 3. Quantum Consciousness Emerging from the EM Field Surrounding Neurons

As discussed above, the Orch OR theory suggests that information processing in the brain occurs at the level of microtubules, which shape neurons and give them their unique architecture. This idea, however, conflicts with established principles of neuroscience. Yet, certain phenomena, such as the binding problem, remain challenging to explain at the scale of neural networks. One idea proposed to address the integration of distinct information processed by localized neural networks across distant regions of the brain—information distributed over a wide spatial area—is to conceptualize consciousness as a force field [68]. Some earlier interpretations [69,70,71] view this force field as having a metaphysical origin, aligning with traditional mind-body dualism. Alternatively, others [23,24,25,26,27,72,73,74] suggest that it represents the brain’s endogenous electromagnetic (EM) field, reframing the mind-body problem as a matter-field dualism.

The Conscious Electromagnetic Information (CEMI) Field Theory, proposed by Johnjoe McFadden [23], serves as the foundation for examining the experimentally observed correlations between synchronized neuronal firing and conscious awareness [75]. According to this theory [24,25,26,27], information processed in local neural networks can be transferred to the brain’s EM field, creating disturbances that reflect this information. It suggests that the information integrated into the EM field corresponds to conscious experience, which can then be re-downloaded into neural networks, influencing the firing patterns of motor neurons. A schematic representation of the CEMI field theory is shown in Figure 2.

The CEMI field theory asserts that synchronized neural firing is essential for transferring information generated by neural computation into the brain’s EM field. Without this synchronization, the theory argues, the brain’s EM field would not be sufficiently perturbed, and the resulting information would contribute only to unconscious awareness.

Moreover, the theory does not claim that integrated information within the EM field directly activates resting neurons. Instead, it suggests that mechanisms, such as the regulation of voltage-gated ion channels—especially when the membrane potential is near the threshold, either slightly below or above—might allow the EM field to influence neuronal activity. This influence could involve either triggering firings that are on the brink of occurring or suppressing those that are barely underway. The theory views this process as a possible basis for free will.

While quantum effects are not considered a requirement, the theory proposes that the EM field could, under specific conditions, control neurons through single-photon interactions. This interaction, it suggests, may represent the only way quantum effects are permitted to manifest within the brain.

The initial studies [24,25] aimed to generate testable predictions by presenting theoretical considerations and experimental evidence related to the origin of the brain’s EM field, the spatiotemporal complexity of its magnitude, and potential mechanisms for interaction with neural computation. Subsequent updates to the theory have further strengthened this proposal. For example, Refs. [26,76,77] reported experiments demonstrating that synchronous neuronal firing has a functional role in the brain and that the brain’s endogenous EM field contributes to recruiting neurons into synchronously firing networks. Additionally, Ref. [27] addressed criticisms suggesting that the binding problem is merely an illusion, showing instead how complex information is unified into coherent ideas that provide meaning within the brain. In another update [78,79,80], it has been proposed that spatially distributed information can only be integrated through energetic fields, such as the EM field.

Despite being based on a solid conceptual foundation, the CEMI field theory has not garnered the attention it warrants in the literature [79]. One contributing factor is the challenges associated with transferring information between the field and matter. While research on the influence of electromagnetic fields on neuronal ion channels is expanding [76], further studies are needed to demonstrate the feasibility of this information transfer. Another reason for the theory’s limited reception is that it still lacks a mathematical model to validate its viability. This limitation is not unique to McFadden’s theory; similar electromagnetic field-based theories of consciousness proposed by Susan Pockett [72,74] and E. Roy John [73] also do not emphasize formal mathematical modeling. Rather, these theories prioritize phenomenological and empirical frameworks over detailed mathematical structures. Despite this, they each examine the role of electromagnetic fields in consciousness in slightly different ways.

Susan Pockett’s work [72,74] highlights the holistic nature of the brain’s electromagnetic activity, suggesting that consciousness emerges not from specific neural circuits but from the global EM field of the brain itself. While McFadden’s theory adopts a more mechanistic approach, positing that synchronized neuronal firing directly affects the brain’s EM field, Pockett’s perspective centers on how the EM field contributes to subjective experience in its entirety. Both theories recognize the essential role of EM fields in consciousness, though they differ in terms of granularity, with McFadden emphasizing neural synchronization and Pockett advocating for a more unified field approach.

Similarly, E. Roy John’s research [73], while acknowledging the importance of electromagnetic fields, focuses on the specific electrical patterns of brain activity. His work highlights the correlation between brain wave patterns and consciousness, suggesting that changes in electrical activity are integral to conscious states. In contrast to McFadden’s emphasis on synchronized neuronal firing as a mechanism for transferring information to the EM field, John’s work explores the relationship between neural oscillations and consciousness more directly, focusing on measurable electrical signals rather than the broader electromagnetic field.

Unlike these classical electromagnetic models, Jibu and Yasue propose a quantum field theory of consciousness [81,82,83], which posits that quantum coherence in the brain’s water molecules, particularly within microtubules, plays a critical role in cognitive processes. Their model suggests that a Bose–Einstein condensation of quantum fields in the brain enables a nonlocal and unified conscious experience. Though speculative, this quantum approach marks a significant departure from classical electromagnetic theories by incorporating quantum field dynamics as a foundational element of consciousness.

### Synchronized Firing Through the Correlations Between Neurons

The central premise of CEMI field theory is that it is not the number of neurons firing, but rather the degree of synchronization in their firing that is related to conscious awareness. On the other hand, synchronization is one of the most important phenomena in the literature on dynamic systems, and there is a significant body of knowledge on the mathematical methods used to study it. However, the complexity of the human brain and the need for dynamic system analysis at the macroscopic level may pose challenges in enriching the CEMI field theory with mathematical models. At this point, information-theoretic approaches may provide an alternative route for both theoretical and experimental research.

There is a natural relationship between the concepts of synchronization and correlation. Strong synchronization in a neural network means that there is a high correlation between individual nerve cells. In other words, information theory could allow us to focus on the individual nerve cells within the brain’s structure, rather than looking at the entire, vast, and complex brain. Moreover, the flow of information within a system can be studied more effectively through the concept of correlation rather than synchronization.

For example, we could model neuronal coherence by developing mathematical models to simulate how neural firing can create coherent EM fields. This can help us understand how these fields might exhibit quantum coherence properties. In this context, existing methods related to the transformations between quantum coherence and correlations in Refs. [84,85,86] can be guiding.

Another approach is to simulate quantum entanglement to explore potential interactions between different regions of the brain. The generation of correlations between massive particles interacting with a gravitational field has been a topic of mathematical investigation, with various experimental proposals investigating whether such interactions could induce entanglement, thereby providing a means to test the quantum nature of the gravitational field [87,88,89]. Moreover, master equation-based approaches demonstrate that fluctuation-mediated Casimir–Polder interactions can lead to correlations that persist, even in the steady state, between initially uncorrelated and spatially separated particles [90]. Following a similar framework, we could model how a specific type of magnetic field might correlate two nerve cells separated by a particular distance. Building upon this, one could design experiments to investigate whether the brain’s electromagnetic field functions in a classical manner, as proposed by McFadden’s CEMI field theory, or whether it exhibits quantum properties, as suggested by Jibu and Yasue.

If synchronized firing produces a classical electromagnetic field, this field would likely create only classical correlations between distant neurons. In contrast, a quantum electromagnetic field could generate quantum coherence or entanglement, potentially involving the brain’s microtubules or water molecules. Quantum information theory provides a robust framework for quantifying both classical and quantum correlations, allowing us to distinguish between these two distinct types of processes. These correlations may correspond to different types of resource management in the brain, with the classical electromagnetic field reflecting local neural synchronization, while the quantum electromagnetic field could be associated with global, nonlocal resource management linked to quantum coherence. Using quantum field theory, we could model how these distinct fields interact and influence neural networks, providing insights into how macroscopic electromagnetic fields may arise from microscopic quantum processes.

These avenues show promise, though they may be computationally complex. Nevertheless, they represent valuable paths for furthering our understanding of electromagnetic field-based theories of consciousness.

## 4. Quantum Consciousness Emerging from the Molecular Interactions Among Neurons

Matthew Fisher proposes a comparison between quantum computing and our brain, suggesting that the latter might function like a quantum computer [28,91]. This model aligns closely with neuroscience conventions, where consciousness is associated with the network of neurocells.

To facilitate quantum computing in the brain, we first need to identify a suitable candidate to act as a qubit. Additionally, the brain’s mechanism should meet the following criteria:Possess a long nuclear-spin coherence time to function as a qubit.Have a method for transporting this qubit throughout the brain and into neurons.Include a molecular scale quantum memory for storing the qubits.Contain a mechanism for quantum entangling multiple qubits.Initiate a chemical reaction that triggers quantum measurements, which in turn determine subsequent neuron firing rates, among other things.

The optimal candidate for a neural qubit in our brain should have a nuclear spin of $\frac{1}{2}$ to ensure long-lived coherence. This is because nuclei with $I>\frac{1}{2}$ experience significant decoherence due to electric field interactions, whereas $I=\frac{1}{2}$ spins are primarily affected by weaker magnetic field perturbations, allowing for much longer coherence times [28]. Therefore, phosphorus emerges as the best candidate for a neural qubit.

The phosphate ion and the pyrophosphate ion serve as transporters for this qubit. In our body, adenosine triphosphate (ATP) undergoes a hydrolysis reaction,$$ATP⟶AMP+PP_{i},$$
producing adenosine monophosphate (AMP) and the pyrophosphate ion (PP*_i_*). Occasionally, further hydrolysis results in the production of two separate phosphate ions.

Once the transporter is identified, a mechanism is needed to protect the coherence of the phosphorus qubit. This is where the Posner molecule comes into play. The existence of stable calcium-phosphate molecules in our body indicates that phosphate ions will bind to calcium cations, protecting them from proton binding. This mechanism extends the coherence time for phosphorus nuclear spin. The calcium-phosphate molecule is referred to as the Posner cluster.

Next, we need a mechanism to entangle two qubits present in two different Posner clusters. The enzyme pyrophosphatase ensures this entanglement. In Posner clusters, pseudospin states arise from the collective nuclear spins of phosphorus atoms. These pseudospin states serve as the quantum degrees of freedom, making the Posner molecule a qutrit due to its three-fold symmetry.

When two Posner molecules attempt to bind, their pseudospin states become entangled, regardless of whether the binding is successful. This entanglement is crucial for quantum computations. The two entangled phosphorus nuclear spins, located in different Posner clusters, influence the firing of corresponding neurons during binding reactions. This means that the reactions in two different neurons are entangled. The schematic showing the two phosphates entangled in two different Posner clusters is provided in Figure 3.

Furthermore, experiments on Posner clusters have been conducted to provide insights into their properties, but their coherence has not yet been studied experimentally [92,93,94,95,96]. Theoretical studies have also been conducted to model the system using a Hamiltonian framework and incorporating environmental noise and some quantum information measures such as concurrence. These studies model Posner molecules with six interacting spins, one for each phosphate molecule. The spin Hamiltonian describes these interactions, considering the symmetry of the Posner molecule and examining various configurations. They use concurrence to analyze entanglement within the spin system [97,98,99]. Another study approaches it from a quantum computing and information perspective, interpreting each step in Fisher’s proposal as a quantum computing operation, though it does not involve environmental factors [100].

Some of these studies are skeptical about the possibility of quantum brain processing. However, our goal here is not to definitively prove or disprove this idea but to illustrate how quantum information could aid in these investigations. To tackle this complex model, we introduce a simplified toy model to specifically investigate the behavior of phosphate molecules, focusing on the preservation of entanglement.

## 5. Study of the Entanglement Preservation

In a recent article, we have explored the preservation of quantum coherence in various geometries [29]. We found that the tetrahedral geometry is the best candidate for the protection of quantum coherence in a central spin, which closely resembles the structure of the phosphate molecule. Building upon this work, we now extend our investigation to the preservation of entanglement.

In our approach, we treat each atom within a phosphate molecule as an identical spin having a magnitude of 1/2. In the context of a phosphate molecule, a cluster of five spins consists of a central spin (associated with the phosphorus atom) surrounded by buffer spins (corresponding to oxygen atoms). The buffer spins interact with individual thermal baths, while the central spin remains isolated from direct environmental interactions. Figure 1 from reference [29] illustrates this concept. When exposed to a magnetic field along the *z*-axis, an energy difference arises between the lower state (spin-up) and the upper state (spin-down), allowing for a two-level description. The buffer spins eventually reach thermal equilibrium with the environment at an inverse temperature β. In our study, we analyze the geometry of buffer networks characterized by different arrays of coupling constants, which are either *g* or 0. These configurations can be represented using planar graphs. Specifically, for a given number of buffer spins, *N*, we explore all feasible buffer networks that can be embedded in a plane (Figure 2 from reference [29] illustrates the different planar graphs). We focus on two extreme cases: one where there is no connectivity among the buffer spins, and another where there is maximum connectivity within the buffer spin network.

For the study of entanglement preservation, we consider two separate clusters with entangled central spins. For example, in a tetrahedral geometry, these clusters could represent two separate phosphate molecules. In each phosphate molecule, the central spin corresponds to the phosphorus atom, and these phosphorus atoms are entangled with each other. Our main goal is to preserve the initial entanglement between two spins, each in a separate network, over time. To accomplish this, we model the system as an open system. The central spins in both networks are entangled with each other, while the surrounding buffer spins remain in a thermal state. There is no direct coupling between the spins of the two networks, as they are well separated. We consider two clusters of N+1 spins, each cluster consisting of a central spin surrounded by a buffer network. The initial cluster state can be represented as a product state,(7)$$\rho \left(0\right)=|\psi_{1}\rangle \langle \psi_{1}|⊗\rho_{th}^{⊗2N},$$
with(8)$$|\psi_{1}\rangle =\frac{|01\rangle +|10\rangle }{\sqrt{2}}.$$
Hereon, we take *ℏ*=1. The thermal state of the buffer spins is defined as(9)$$\rho_{th}=\frac{e^{-\beta \frac{\omega }{2}{\hat{\sigma }}_{z}}}{Z},$$
where $\sigma_{z}=|0\rangle \langle 0|-|1\rangle \langle 1|$ represents the Pauli-*z* operator and Z=2cosh[βω/2] denotes the partition function. Here, we assume the Boltzmann constant $k_{B}$ to be equal to 1. The total Hamiltonian of one spin cluster reads as(10)$$\hat{H}=\sum_{i=1}^{N+1}\frac{\omega }{2}{\hat{\sigma }}_{z}^{\left(i\right)}+\sum_{i\neq j}g_{ij}\left({\hat{\sigma }}_{x}^{\left(i\right)}{\hat{\sigma }}_{x}^{\left(j\right)}+{\hat{\sigma }}_{y}^{\left(i\right)}{\hat{\sigma }}_{y}^{\left(j\right)}\right)\;.$$
The Pauli spin-1/2 operators for the $i^{th}$ spin are denoted by $\sigma_{x}^{\left(i\right)}$, $\sigma_{y}^{\left(i\right)}$, and $\sigma_{z}^{\left(i\right)}$, and the interaction strength between the spin pair (i,j) is represented by $g_{ij}$. The central spin, labeled by “1”, is coupled to the buffer spins at strength $g_{1j}=g\neq 0$ in all geometries under consideration. On the other hand, each buffer network is defined by a different array of coupling constants consisting of zero or *g* values, which can be represented by a planar graph. In other words, we consider two different geometries of the spins in the buffer network: either they do not interact with each other and are coupled only to the central spin (forming a topology that resembles a *star graph*), or they are fully connected to all the other spins in the system (giving rise to a *complete graph*). To be precise, for N=5, the fully connected geometry is not a complete graph, as the central spin is positioned right in the middle of the bulk network (more details are given in [29]), hindering the interaction between spin 2 and spin 6, which are uncoupled.

We describe the open quantum system dynamics of the spin network by the following Lindblad master equation [101],(11)$$\dot{\rho }\left(t\right)=-i\left[\hat{H},\rho \left(t\right)\right]+D\left(\rho \left(t\right)\right)\;,$$
where the unitary contribution to the dynamics is provided by the self-Hamiltonian of the system given in Equation (Equation 10). By assuming weakly coupled buffer spins, local thermal dissipation channels are described by the dissipator in Equation (Equation 11) [102,103,104],(12)$$\begin{array}{c} \begin{array}{cc} D(\rho ) & =\sum_{i=2}^{N+1}\gamma_{i}\;\left(1+n\left(\omega \right)\right)\left[{\hat{\sigma }}_{i}^{-}\rho \left(t\right){\hat{\sigma }}_{i}^{+}-\frac{1}{2}\left\{{\hat{\sigma }}_{i}^{+}{\hat{\sigma }}_{i}^{-},\rho \left(t\right)\right\}\right]\\ & +\sum_{i=2}^{N+1}\gamma_{i}\;n\left(\omega \right)\;\left[{\hat{\sigma }}_{i}^{+}\rho \left(t\right){\hat{\sigma }}_{i}^{-}-\frac{1}{2}\left\{{\hat{\sigma }}_{i}^{-}{\hat{\sigma }}_{i}^{+},\rho \left(t\right)\right\}\right]\;, \end{array} \end{array}$$
where n(ω) is the Planck distribution at the spin resonance frequency ω, ${\hat{\sigma }}^{\pm }$ are the Pauli spin ladder operators, and $\gamma_{i}=\gamma$ is the coupling constant between the environment and the $i^{th}$ buffer spin, taken to be homogeneous for each buffer spin independently of the network structure for simplicity.

To verify the entanglement protection of different geometries we look at the logarithmic negativity. The negativity, denoted as N(ρ), quantifies entanglement in bipartite quantum systems, $\rho_{AB}$ [105]. It is given by summing over the absolute of the negative eigenvalues, $\lambda_{i}$, of the partially transposed density matrix, $\rho_{A}^{T}$,(13)$$N\left(\rho_{AB}\right)=\sum_{i}\left|\lambda_{i}\right|.$$
The logarithmic negativity, $E_{N}$, is related to the negativity and serves as a good indicator of the degree of entanglement. It is defined as(14)$$E_{N}\left(\rho_{AB}\right)=\log_{2}\left(2N\left(\rho_{AB}\right)+1\right).$$
Notably, Figure 4 illustrates the logarithmic negativity over time between the two central spins in different geometries with interaction between buffer spins. We observe that entanglement is conserved for an extended duration in the case of N=4 with maximum connectivity between buffer spins as shown in Table 1. This geometry is the closest to the tetrahedral geometry of the phosphate molecule. We have also compared the protection time of entanglement between vanishing and maximal connectivities in the buffer network and shown that protection is optimal when interactions are turned on. In our previous work, we provided physical insight into the heat transfer within the system, noting that a delay is particularly pronounced in tetrahedral geometries. This explanation suggests that a similar mechanism could apply to the preservation of entanglement. The initial calculations yield promising results, offering insights into the feasibility of Fisher’s proposal. This marks the first step toward developing a more realistic model, where the spins more accurately mirror the biological environment. In the following section, we use a sectional diagonalization of the Hamiltonian to more clearly reveal the effect of buffer isolation on quantum information protection.

## 6. Theoretical Explanation

### 6.1. Hamiltonian Transformation

We now examine the system with a fully connected buffer network, and more specifically its Hamiltonian, from a different perspective, in an attempt to provide a theoretical explanation of better entanglement preservation in the fully connected geometry. By applying a unitary transformation to the Hamiltonian, we can transition from a fully connected graph of buffer spins to the other extreme, where the buffer spins do not interact with each other. These fictitious non-interacting buffer spins will be referred to as the dressed buffer spins and the new representation after the transformation as the dressed picture.

We rewrite the Hamiltonian in Equation (Equation 10) in terms of ladder operators as(15)$$\begin{array}{cc} \hat{H} & =\sum_{i=1}^{N+1}\frac{\omega }{2}\left(\sigma_{i}^{+}\sigma_{i}^{-}-\sigma_{i}^{-}\sigma_{i}^{+}\right)+\sum_{i\neq j}2g_{ij}\left(\sigma_{i}^{+}\sigma_{j}^{-}+\sigma_{i}^{-}\sigma_{j}^{+}\right) \end{array}$$(16)$$\begin{array}{cc} \; & =\sum_{i=1}^{N+1}\frac{\omega }{2}\left(2\sigma_{i}^{+}\sigma_{i}^{-}-1^{\left(i\right)}\right)+\sum_{i\neq j}2g_{ij}\left(\sigma_{i}^{+}\sigma_{j}^{-}+\sigma_{i}^{-}\sigma_{j}^{+}\right). \end{array}$$
Given the spin operator vectors,(17)$$\begin{array}{cc} \Sigma_{+} & =(\sigma_{1}^{+},\ldots ,\sigma_{N}^{+}), \end{array}$$(18)$$\begin{array}{cc} \Sigma_{-} & =(\sigma_{1}^{-},\ldots ,\sigma_{N}^{-}), \end{array}$$
the Hamiltonian, in matrix form, can be expressed as (up to an irrelevant constant):(19)$$\hat{H}=\Sigma_{+}H\Sigma_{-}^{T},$$
where *H* is given by(20)$$H=\left[\begin{array}{cc} \omega & v\\ inev^{T} & H^{\prime } \end{array}\right].$$
The vector *v* represents the interaction between the central spin (spin 1) and the other spins *j* (for j=2,…,N+1), expressed as(21)$$v=\left(2g_{12},\ldots ,2g_{1N+1}\right)=\left(G,...,G\right),\;\;\mathrm{with}\;G=2g.$$
Moreover, note that, for N=2,3,4, $H^{\prime }=\omega 1+GA_{\mathrm{com}}$, where $A_{\mathrm{com}}$ is the adjacency matrix of the complete graph of order *N* up to a sign.

Here, we aim to preserve the first spin, corresponding to the central spin, while diagonalizing $H^{\prime }$,(22)$$H^{\prime }=UDU^{†}.$$
For N=2,3,4, this is equivalent to finding the eigenvalues and eigenvectors of the complete graph $A_{\mathrm{com}}u_{j}=\mu_{j}u_{j}$. We readily obtain $\mu_{1}=\ldots =\mu_{N-1}=-1$ with multiplicity N−1, and $\mu_{N}=N-1$ with multiplicity 1. The corresponding eigenvectors are $u_{j}=\frac{1}{\sqrt{2}}\left(|N\rangle -|j\rangle \right)$ for j=1,…,N−1, and $u_{N}=\frac{1}{\sqrt{N}}\sum_{j=1}^{N}|j\rangle .$ The matrix *D* in (Equation 22) is diagonal and contains the frequencies of the dressed buffer spins, which, following our discussion on the eigenvalues of the complete graph and some trivial algebra for the geometry with N=5, are given by:for N=2,(23)$$\left({\tilde{\omega }}_{2},{\tilde{\omega }}_{3}\right)=\left(\omega +2g,\omega -2g\right);$$for N=3,(24)$$\left({\tilde{\omega }}_{2},{\tilde{\omega }}_{3},{\tilde{\omega }}_{4}\right)=\left(\omega +4g,\omega -2g,\omega -2g\right);$$for N=4,(25)$$\left({\tilde{\omega }}_{2},{\tilde{\omega }}_{3},{\tilde{\omega }}_{4},{\tilde{\omega }}_{5}\right)=\left(\omega +6g,\omega -2g,\omega -2g,\omega -2g\right);$$for N=5,(26)$$\left({\tilde{\omega }}_{2},{\tilde{\omega }}_{3},{\tilde{\omega }}_{4},{\tilde{\omega }}_{5},{\tilde{\omega }}_{6}\right)=\left(\omega +2\left(1+\sqrt{7}\right)g,\omega +2\left(1-\sqrt{7}\right)g,\omega ,\omega -2g,\omega -2g\right).$$
The spin operator vectors are modified in the dressed picture as $\Sigma \to \tilde{\Sigma }$ with(27)$$\tilde{\Sigma }=\left(1⊗U^{†}\right)\Sigma ,$$
which gives(28)$$\tilde{\Sigma }=\left(\sigma_{1},{\tilde{\sigma }}_{2},\ldots ,{\tilde{\sigma }}_{N}\right).$$
As discussed before, $\sigma_{1}$, which represents the spin operator of the central spin, remains unchanged.

Next, we rewrite the interaction between the central and the buffer spins in the dressed picture $\tilde{\Sigma }$, making use of the transformation,(29)$$\sigma_{i}^{\pm }=\sum_{j=2}^{N+1}U_{ij}{\tilde{\sigma_{j}}}^{\pm }.$$
We can thus define a new vector of interactions $\tilde{v}=\left({\tilde{g}}_{12},\ldots ,{\tilde{g}}_{1N+1}\right)$. Note, however, that the coupling between the central spin and the buffer spins can be written as $G\sigma_{1}^{+}\sum_{j=2}^{N+1}\sigma_{j}^{-}+H.c.$, and in the case of a complete graph the sum $\sum_{j=2}^{N+1}\sigma_{j}^{-}$ corresponds to a single eigenvector of the graph ($u_{N}$ defined above, with eigenvalue N−1). In other words, for N=2,3,4, the central spin is coupled to a single dressed buffer spin with energy ω+(N−1)G. Furthermore, the coupling constant is equal to $G\sqrt{N-1}$. Finally, for the case with N=5, the buffer spins do not form a perfectly complete graph, as the interaction between spin 2 and spin 6 is missing. However, a similar reasoning applies and it can be shown that the central spin is coupled to only two dressed buffer spins, as depicted in Figure 5 (column C). We remark that the decoupling of the central spin from most of the dressed buffer spins is due to the system Hamiltonian being highly symmetric.

Let us finally address the coupling of the buffer spins with the local baths in the dressed picture. The local thermal dissipation channels are described by the dissipator (Equation 12). By applying the transformation (Equation 29), we obtain in the dressed picture the same dissipator with $\sigma_{i}$ replaced by the suitable linear combination of ${\tilde{\sigma }}_{j}$. In the dressed picture, the separate baths acting locally on each buffer spins become global baths [102,103] acting collectively on the dressed buffer spins.

In conclusion, through the change of variables (Equation 29) we have mapped the problem of a central spin coupled to a network of fully connected buffer spins to the problem of a central spin coupled to a collection of non-interacting fictitious buffer spins, which resembles the other model we consider in this work that displays worse entanglement preservation (see for instance Table 1). Contrary to the latter scenario, however, only one of the dressed buffer spins is coupled to the central spin, although through a stronger interaction and under the action of global baths. This may give us some hints of why the fully connected geometry better preserves entanglement, as we show in the following.

### 6.2. Calculation for N=2

Let us explicitly show the calculation for N=2. For N=2,(30)$$D=\left[\begin{array}{cc} \omega +2g & 0\\ 0 & \omega -2g \end{array}\right].$$
The transformation matrix,(31)$$U=\frac{1}{\sqrt{2}}\left[\begin{array}{cc} 1 & -1\\ 1 & 1 \end{array}\right],$$
transforms the spin matrices as(32)$$\begin{array}{cc} \sigma_{2}^{\pm } & =\frac{1}{\sqrt{2}}\left({\tilde{\sigma }}_{2}^{\pm }-{\tilde{\sigma }}_{3}^{\pm }\right); \end{array}$$(33)$$\begin{array}{cc} \sigma_{3}^{\pm } & =\frac{1}{\sqrt{2}}\left({\tilde{\sigma }}_{2}^{\pm }+{\tilde{\sigma }}_{3}^{\pm }\right). \end{array}$$
Then, the interaction of the first spin with the buffer spins is transformed as(34)$$G\left(\sigma_{1}^{+}\sigma_{2}^{-}+\sigma_{1}^{+}\sigma_{3}^{-}\right)⟶\frac{2}{\sqrt{2}}G\left(\sigma_{1}^{+}{\tilde{\sigma }}_{2}^{-}\right),$$
with ${\tilde{g}}_{12}=\sqrt{2}G$. It is worth noting that after diagonalization, the interactions between dressed buffer spins are suppressed.

### 6.3. Interpretation of the Results

In the non-interacting case, there are *N* interactions between the central spin and buffer spins (column A in Figure 5). When interactions between buffer spins are introduced (column B in Figure 5), the interaction between the central spin and the dressed buffer spins is reduced to a single interaction for N=2 to N=4, and to two interactions for N=5 (as shown in column C in Figure 5). Using the dressed picture, we can more easily compare these scenarios and understand why interacting buffer spins offer better quantum information protection. In column B, where the buffer spins are maximally interacting, the interaction between the central spin and the buffer network (which is connected to the environment and responsible for the loss of quantum information) is significantly modified. While in the case of non-interacting buffer spins (column A) the central spin is weakly coupled to *N* independent and locally dissipating buffer spins, if the buffer network is fully connected (N=2,3,4), then the central spin interacts with a single dressed buffer spin (column C), which is dissipating into different global baths in a more and more intense way as a function of *N* (coupling constant going as $\sqrt{N}$).

Focusing now on the dynamics of the central spin only, the difference between the two scenarios described above lies in a transition from a Markovian to a non-Markovian dynamics. Indeed, if dissipation comes from a collection of identical and weakly coupled systems, the dynamics have a more Markovian behavior corresponding to a roughly flat spectral density of the environment (this is an exact result if we consider N→∞ non-interacting buffer spins [101]). On the other hand, a stronger coupling with a single dressed buffer spin implies a more and more non-Markovian behavior, as the dressed spin filters the thermal white noise into Lorentzian noise. Furthermore, it is well-known that non-Markovian dynamics better preserve quantum information, as non-Markovianity induces a backflow of quantum information from the reservoir to the central spin [106,107,108,109,110,111,112,113,114,115,116]. It is thus reasonable to expect that the fully connected buffer network preserves entanglement during a longer time than the non-interacting network, as we indeed observe.

For N≥5, it is physically impossible to construct a fully connected network, given the position of the central spin (see e.g., [29]), and for instance for N=5 the interaction between spin 2 and spin 6 is missing. So, the dressed buffer network does not reduce to a single interaction. Instead, the central spin is coupled to two non-interacting dressed buffer spins, while three more are left uncoupled. We can draw similar conclusions as for the case of perfectly fully connected buffer network, i.e., the dynamics will have a more non-Markovian behavior than in the case of non-interacting buffer network, leading to better entanglement preservation. This behavior, however, will be less pronounced than in the case of a fully connected network.

Finally, we point out that in the tetrahedral geometry (N=4), there is a significant improvement in quantum information preservation (Figure 4). We observe that N=4 presents the strongest possible coupling between the central spin and a singled dressed buffer spin (Figure 5), as for N≥5 the fully connected geometry is not possible. Consequently, quantum information would be preserved longer due to the strongest possible non-Markovian behavior, arising in the case of N=4.

### 6.4. Numerical Verification

To numerically verify this observation, we can calculate the trace distance measure of non-Markovianity [117]. Trace distance between two density matrices $\rho_{1}\left(t\right)$ and $\rho_{2}\left(t\right)$ at time *t* can be calculated as(35)$$D\left(\rho_{1}\left(t\right),\rho_{2}\left(t\right)\right)=\frac{1}{2}{∥\rho_{1}\left(t\right)-\rho_{2}\left(t\right)∥}_{1},$$
where ${∥A∥}_{1}$ denotes the trace norm of matrix *A*, which is the sum of the singular values of *A*. To verify non-Markovianity, we use the trace distance to compare two initially orthogonal states. The initial states of the central spins, $|\phi_{1}\rangle$ and $|\phi_{2}\rangle$, corresponding to $\rho_{1}$ and $\rho_{2}$ in the trace distance measure, are taken as(36)$$\begin{array}{cc} |\phi_{1}\rangle & =\frac{1}{\sqrt{2}}\left(|0\rangle +|1\rangle \right); \end{array}$$(37)$$\begin{array}{cc} |\phi_{2}\rangle & =\frac{1}{\sqrt{2}}\left(|0\rangle -|1\rangle \right). \end{array}$$
Non-Markovianity is identified when the trace distance between these states, which reflects their distinguishability, increases at certain times during their evolution. This increase indicates a backflow of information from the environment (buffer network) to the system.

After evolving the two initial states, $|\phi_{1}\rangle$ and $|\phi_{2}\rangle$, we then compare them at each time instant using the trace norm distance. As shown in Figure 6, for N=4, we observe more pronounced oscillations, indicating higher non-Markovianity. This may explain our earlier findings and the enhanced protection observed in the tetrahedral geometry.

## 7. Conclusions

While this overview has aimed to summarize the current state of research on the brain and its functions, it cannot fully encompass the breadth and depth of the field. Our focus has been primarily on the biological and cognitive aspects, and we have not delved into the more speculative intersections between brain research and foundational philosophical or physical theories. These areas, though of growing interest, remain largely in the realm of ongoing exploration and debate. The brain, with its complex and still largely mysterious nature, serves as a central link between our perception of the world and our identity. Its intricate functioning underpins much of scientific inquiry, making the study of the brain vital not only for addressing neurological diseases but also for advancing fields like artificial intelligence and even fundamental physics. Recent discussions in the literature suggest that a deeper understanding of brain function may require revisiting foundational questions in quantum physics and relativity. For instance, the role of the observer in quantum experiments remains an open debate, with potential implications for cognition and perception [118,119,120,121]. Some researchers explore whether consciousness could have a physical basis extending beyond biological systems [118,119], while others investigate whether principles from relativity might provide insights into the nature of thought and awareness [122]. Though these questions are speculative and far from settled, they underscore the importance of interdisciplinary approaches in advancing our understanding of the brain and its connections to more fundamental domains of science.

In this perspective, we explored several quantum brain theories described in the literature. We began with the hypothesis of consciousness occurring within microtubules, then moved on to its potential explanation within the electromagnetic field, culminating with an analysis of cognition via the Posner cluster model. Specifically, we provided insights into this model by examining a toy model that resembles the phosphate molecule in the Posner cluster and looking at the preservation of quantum entanglement. Our findings suggest that the tetrahedral geometry, adopted by the phosphate molecule in the Posner Cluster, offers superior entanglement protection. To further elucidate our results and the effects in different geometries, we map the geometry to an alternative representation by diagonalizing the Hamiltonian. This approach more clearly reveals the impact of an effective almost isolated buffer network (a single dressed buffer spin coupled to the central spin) on quantum information protection.

While our findings do not conclusively prove Matthew Fisher’s proposal on cognition, they offer valuable insights into how simplified models and quantum information measures can provide different perspectives and enhance our understanding of these theories.

## Figures and Tables

**Figure 1 entropy-27-00243-f001:**
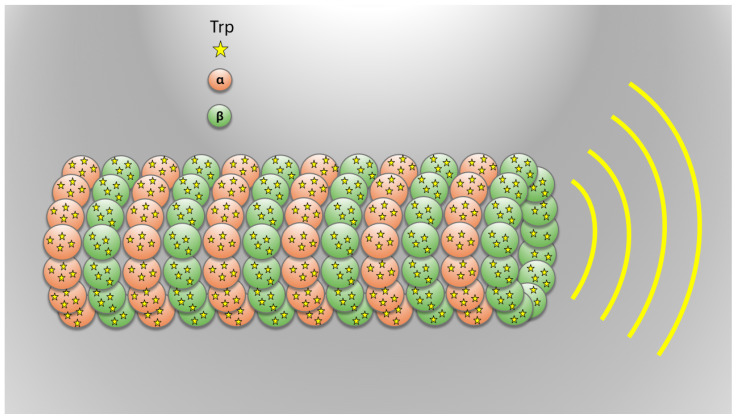
Schematic representation of the cylindrical microtubule structure formed by αβ-tubulin dimers (α in orange, β in green), highlighting the tryptophan (Trp) network. The Trp residues are depicted as stars, illustrating their collective emission of light, a phenomenon associated with superradiance.

**Figure 2 entropy-27-00243-f002:**
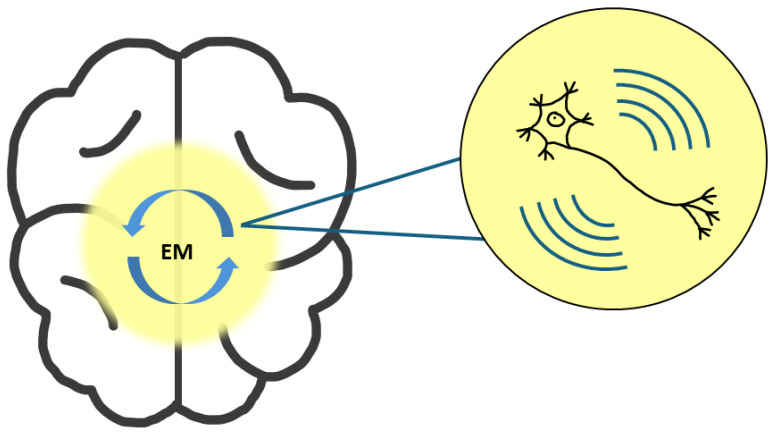
Schematic representation of the CEMI (Conscious Electromagnetic Information) field theory, illustrating how the electromagnetic (EM) field generated by neural networks interacts with and influences neuronal activity.

**Figure 3 entropy-27-00243-f003:**
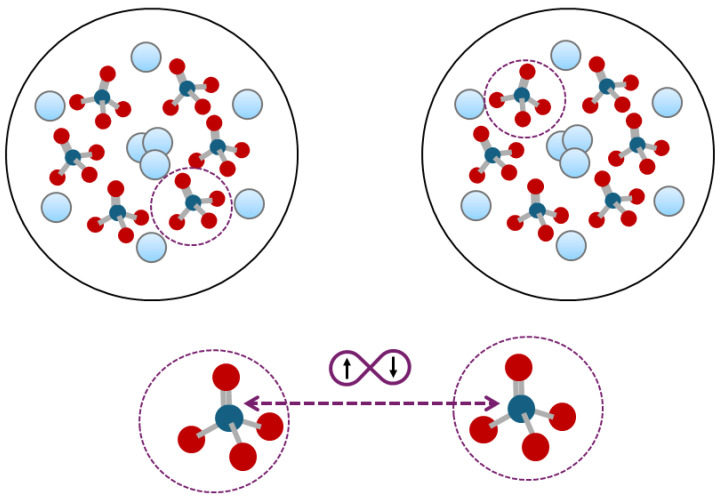
Schematic representation of two Posner clusters, each composed of nine calcium atoms and six phosphate ions (Ca_9_(PO_4_${)}_{6}^{3-}$). The diagram illustrates how two phosphate ions can become entangled between different Posner clusters.

**Figure 4 entropy-27-00243-f004:**
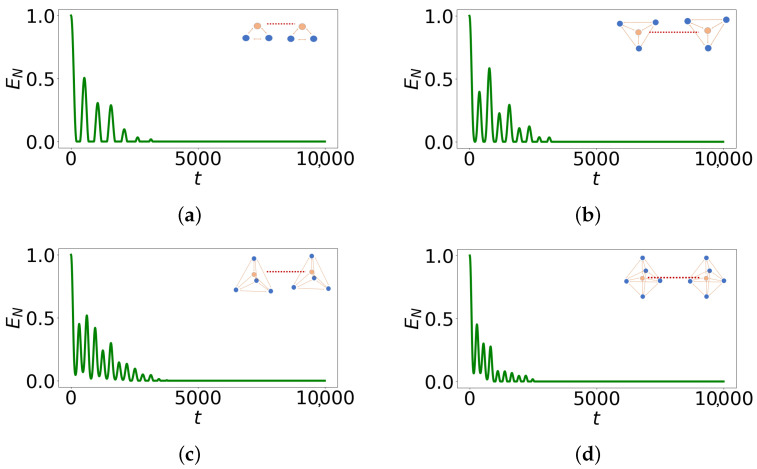
Logarithmic negativity, $E_{N}$, of the reduced state composed on the two entangled spins with respect to dimensionless time, *t*. Here, all the results refer to the model with fully connected buffer spins (for N=5 the interaction between spin 2 and spin 6 is missing). (**a**) N=2. (**b**) N=3. (**c**) N=4. (**d**) N=5.

**Figure 5 entropy-27-00243-f005:**
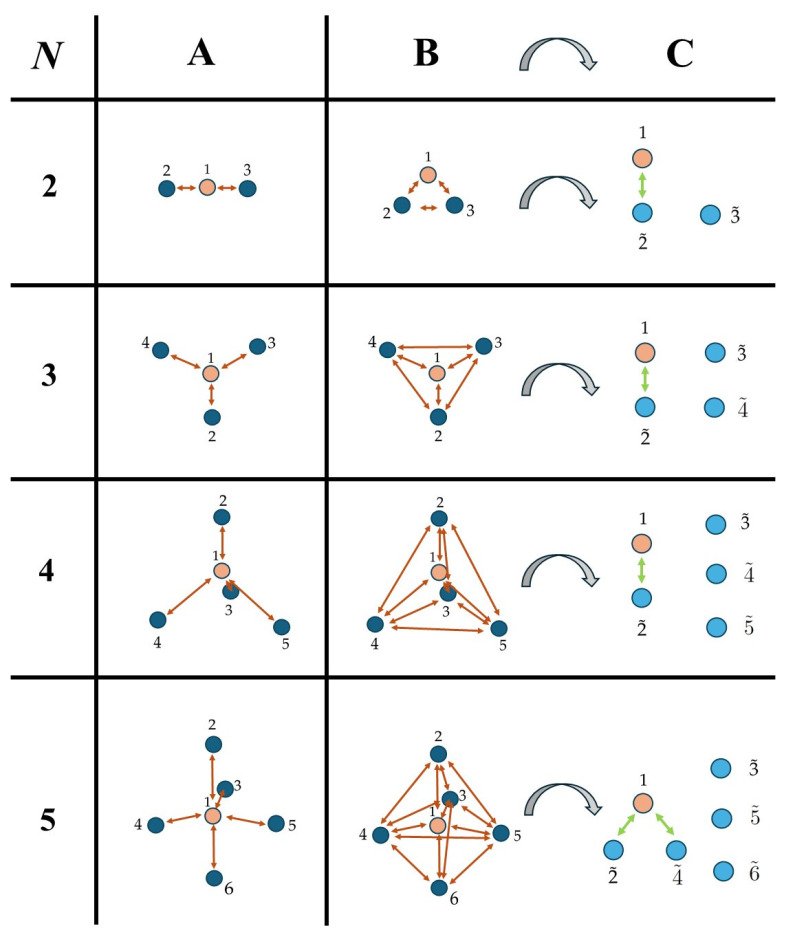
The first column represents the number of buffer spins, *N*. Column A represents the non-interacting buffer spins. Column B shows the interacting buffer spins. Column C shows the dressed buffer spins after the Hamiltonian transformation which is represented by the curved arrows. In the original representation (column B), all spin couplings are identical. In the new representation, that is, the dressed picture (column C), the couplings vary as follows: ${\tilde{g}}_{12}=1.41G$ for N=2, ${\tilde{g}}_{12}=1.73G$ for N=3, ${\tilde{g}}_{12}=2G$ for N=4, and ${\tilde{g}}_{12}=-2.23G$, ${\tilde{g}}_{14}=0.21G$ for N=5.

**Figure 6 entropy-27-00243-f006:**
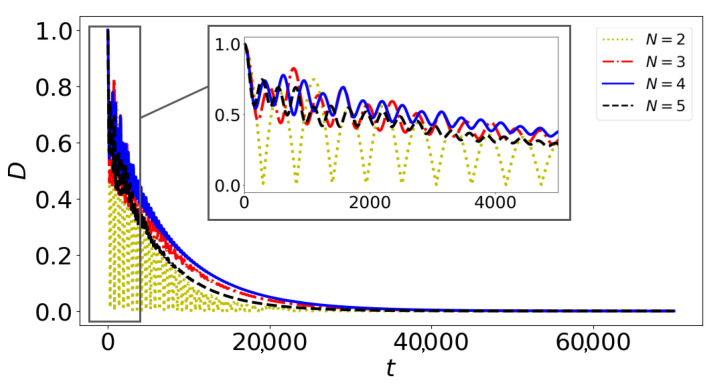
Trace norm distance, *D*, between $\rho_{1}$ and $\rho_{2}$ over dimensionless time for different numbers of buffer spins: N=2, N=3, N=4, and N=5. The buffer spins interact with each other.

**Table 1 entropy-27-00243-t001:** Time, $t_{1}$, at which the logarithmic negativity between the two center spins is lower than $10^{-4}$. The left and right columns correspond to vanishing and maximal connectivities in the buffer network respectively at fixed *N*.

N+1	3		4		5		6	
$t_{1}$	2250	3210	2330	3230	1260	**3800**	1140	2540

## Data Availability

No new data were created or analyzed in this study. Data sharing is not applicable to this article.

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
