# Peer review of "Quantum Models of Consciousness from a Quantum Information Science Perspective"

_entropy, 2025, doi:10.3390/e27030243_

Round 1

Reviewer 1 Report

Comments and Suggestions for Authors

Please, see the attached pdf file.

Reviewer 2 Report

Comments and Suggestions for Authors

This is a well written and well-prepared manuscript that reviews three different theories of cognition and consciousness.  The manuscript also provides some detailed theoretical analyses designed to examine the plausibility of one of the proposed theories.

The authors first review the Penrose  - Hameroff Orch-Or theory. They provide a brief but accurate review of this theory. However, they didn’t provide much discussion about the theory and empirical evidence for length of time for maintaining coherence in microtubules.

Second,  the authors review ideas put forth for electromagnetic field theories of consciousness. As the authors point out, the theory by McFadden was discussed for some length, but similar ideas have been proposed by Pockett and John that should also be mentioned.  As the authors point out, these ideas have not been mathematically developed.  Also, it seems not quite appropriate to call these models quantum. They seem to rely mostly on classical electrical field ideas.

However, in this section the authors could discuss the Jibu and Yasue who proposed a quantum field theory of consciousness.

Third, the authors review Fisher’s theory of quantum cognition produced by entangled ions protected within Posner molecules. The authors mention on page 8 line 328 that there is some experimental evidence for this theory. Unfortunately, the authors didn’t go into much detail here. What is the empirical evidence for Posner molecules protecting decoherence? How long has coherence been actually observed (instead of theoretically predicted) to occur.

The remainder of the article goes into considerable mathematical detail describing theoretical analyses of the Fisher theory to evaluate its plausibility.

In sum, I think this is a very good review but I would like to see

  1. Provide more discussion about the theory and empirical evidence for length of time for maintaining coherence in microtubules.
  2. Discuss the Jibu and Yasue quantum field theory of consciousness
  3. Describe the empirical evidence for Posner molecules protecting decoherence. How long has coherence been actually observed (instead of theoretically predicted) to occur.

Round 2

Reviewer 1 Report

Comments and Suggestions for Authors

Authors have done excellent work to revise their article. It is very informative, useful, and pleasurable to read this review now for the second time. I strongly recommend to publish it immediately, it will surely find its readers.